# Tumor Size Matters—Understanding Concomitant Tumor Immunity in the Context of Hypofractionated Radiotherapy with Immunotherapy

**DOI:** 10.3390/cancers12030714

**Published:** 2020-03-18

**Authors:** Jean Philippe Nesseler, Mi-Heon Lee, Christine Nguyen, Anusha Kalbasi, James W. Sayre, Tahmineh Romero, Philippe Nickers, William H. McBride, Dörthe Schaue

**Affiliations:** 1Department of Radiation Oncology, University of California at Los Angeles (UCLA), Los Angeles, CA 90095-1714, USA; JNesseler@mednet.ucla.edu (J.P.N.); Mi-HeonLee@mednet.ucla.edu (M.-H.L.); christinenguyenx3@yahoo.com (C.N.); AnushaKalbasi@mednet.ucla.edu (A.K.); WMcBride@mednet.ucla.edu (W.H.M.); 2School of Public Health, Biostatistics and Radiology, University of California at Los Angeles (UCLA), Los Angeles, CA 90095-1714, USA; jsayre@ix.netcom.com; 3Department of Medicine Statistics Core University of California at Los Angeles (UCLA), Los Angeles, CA 90095-1714, USA; TahminehRomero@mednet.ucla.edu; 4Department of Radiation Oncology, Centre François Baclesse, Esch-sur-Alzette L-4240, Luxembourg, Luxembourg; philippe.nickers@baclesse.lu

**Keywords:** fibrosarcoma, tumor size, radiation therapy, programmed cell death protein 1

## Abstract

The purpose of this study was to determine the dynamic contributions of different immune cell subsets to primary and abscopal tumor regression after hypofractionated radiation therapy (hRT) and the impact of anti-PD-1 therapy. A bilateral syngeneic FSA1 fibrosarcoma model was used in immunocompetent C3H mice, with delayed inoculation to mimic primary and microscopic disease. The effect of tumor burden on intratumoral and splenic immune cell content was delineated as a prelude to hRT on macroscopic T1 tumors with 3 fractions of 8 Gy while microscopic T2 tumors were left untreated. This was performed with and without systemic anti-PD-1. Immune profiles within T1 and T2 tumors and in spleen changed drastically with tumor burden in untreated mice with infiltrating CD4+ content declining, while the proportion of CD4+ Tregs rose. Myeloid cell representation escalated in larger tumors, resulting in major decreases in the lymphoid:myeloid ratios. In general, activation of Tregs and myeloid-derived suppressor cells allow immunogenic tumors to grow, although their relative contributions change with time. The evidence suggests that primary T1 tumors self-regulate their immune content depending on their size and this can influence the lymphoid compartment of T2 tumors, especially with respect to Tregs. Tumor burden is a major confounding factor in immune analysis that has to be taken into consideration in experimental models and in the clinic. hRT caused complete local regression of primary tumors, which was accompanied by heavy infiltration of CD8+ T cells activated to express IFN-γ and PD-1; while certain myeloid populations diminished. In spite of this active infiltrate, primary hRT failed to generate the systemic conditions required to cause abscopal regression of unirradiated microscopic tumors unless PD-1 blockade, which on its own was ineffective, was added to the RT regimen. The combination further increased local and systemically activated CD8+ T cells, but few other changes. This study emphasizes the subtle interplay between the immune system and tumors as they grow and how difficult it is for local RT, which can generate a local immune response that may help with primary tumor regression, to overcome the systemic barriers that are generated so as to effect immune regression of even small abscopal lesions.

## 1. Introduction

Recent clinical immunotherapy (IT) trials have demonstrated that immune checkpoint inhibitors (ICI) can cause immune rejection of a minority of human tumors [1]. A prerequisite for ICI efficacy is a state of preexisting tumor immunity that is often correlated with intratumoral CD8+ T lymphocyte infiltration [2], modified by many other conditions and factors such as the PD-1/PD-L1 pathway. Cytotoxic CD8+ T cells are almost always a critical element driving tumor immune regression, for which tumor regression antigens as initiators and targets are prerequisites. This also means that generating de novo tumor immunity is of paramount importance if the scope of successful IT is to be broadened. The possibility of integrating ICIs within radiation therapy (RT) of solid cancers is being intensely investigated in what can only be described an unprecedented number of clinical trials [3], but the efficacy of this combined approach is still uncertain. Furthermore, there are many features of the interaction of RT with the immune system in cancer therapy that need further elucidation.

We and others have shown that RT can boost anti-tumor immune responses clinically [4,5,6], but rarely does RT convert a non-responder into a responder. Similar observations can be obtained in preclinical models of IT-RT combinations. Immunogenic murine tumors, such as the FSA1 fibrosarcoma used in this study, grow relentlessly in the face of concomitant immunity, defined as a state of effective immunity coupled with progressive tumor growth. Numerous tumor immune escape mechanisms have been proposed. For this tumor model, we have previously described two-zones of tolerance that are generated by small and large tumor burdens, with an intermediate window of concomitant immunity [7]. In this model, as tumors grow, tumor-specific Tregs are the first to downregulate immunity and this is followed later by non-specific immune suppression by myeloid cells. Under some circumstances, removal of the primary by RT, or surgery, has been reported to unmask an immune state [8]. Since concomitant immunity can decrease the radiation dose required for primary tumor cure [9], understanding how immunosuppression functions is essential to improving local and distant radiation effects. Furthermore, if immunity were to be effective against microscopic metastatic disease, a compelling case can be made for treating oligometastatic disease. The importance of tumor burden in driving immunosuppression cannot be overstated, however it remains a largely unrecognized confounding factor when assessing tumor immunity in response to RT, with and without IT. 

Here, we use a moderately immunogenic murine model of fibrosarcoma to explore the effect of hypofractionated RT (hRT) with and without PD-1 blockade on antitumor immunity as it relates to tumor load and examine the ability of the concomitant immune state to deal with distant microscopic disease. Clinically, localized high-risk soft tissue sarcoma (STS) is typically treated with wide local resection in combination which RT in the neoadjuvant or adjuvant setting with or without chemotherapy [10,11] but distant recurrences remain a major cause of mortality suggesting the presence of micrometastatic disease at the time of initial treatment [12]. Neoadjuvant hRT is under investigation as a novel treatment approach to STS, as is anti-PD-1 treatment, which speaks to the relevance of the study.

## 2. Results

### 2.1. Tumor Load Determines the Immune Balance—Locally and Systemically

Immunogenic FSA1 tumors grow progressively [13] as concomitant immunity becomes suppressed, first by tumor-specific Tregs and later by non-specific activated myeloid cells [7,14]. Less is known about how therapy-induced tumor regression influences the immune system locally and systemically [8]. As a prelude to such studies, in a series of experiments we examined how immune parameters in untreated T1 and T2 tumors and in spleen were influenced by tumor burden (Figure 1). In this model, a second inoculum (T2) was injected into the contralateral side 7 days after the primary injection. Not surprisingly, not all T2 inocula grew indicating a state of immunity from the T1 injection, but some did yield progressively growing tumors (Figure 2B). The data from these experiments were pooled and intratumoral lymphoid and myeloid cell populations in T1 and T2 tumors were plotted against individual tumor size or, for the spleen, against combined size (T1 + T2) in individual linear regression graphs and summarized in heatmaps (Figure 1D–F, Figure 2).

In the lymphoid compartment (Figure 1A), as T1 and T2 tumors grew their content of CD4+ lymphocytes decreased progressively (T1, R = −0.493, *p* = 0.027), while the fraction of regulatory CD4+ T cells (Tregs) increased (T1, R = 0.77, *p* < 0.001). This scenario was mirrored in the spleen (CD4, R = −0.485, *p* = 0.03; Tregs, R = 0.715, *p* < 0.001). The CD8+ cell compartment also tended to shrink, but this was significant only in the spleen (T1, n.s.; spleen, R = −0.487, *p* = 0.029). Their activation status, assessed by PD-1 expression and intracellular IFN-γ levels, were little affected. 

The myeloid compartment was more variable between locations (Figure 1B) with CD11b cells increasing in T1, but not T2 tumors and spleen. Polymorphonuclear-myeloid derived suppressor cells (PMN-MDSC) increased only in larger T1 tumors (>10 mm diameter, R = 0.674, *p* = 0.001) while M2 macrophages showed more measured increases in T1 tumors (T1, R = 0.695, *p* = 0.001). The percent of M-MDSC and PD-L1+ CD11b+ myeloid cells increased in the spleen (R = 0.564, *p* = 0.01; R = 0.604, *p* = 0.005, respectively), but not in tumors although values in tumors were generally high, perhaps leaving little room for further increases. PD-L1 expression levels did rise in T1 tumors and spleens, suggesting activation. Overall, the myeloid content of T2 tumors was hardly affected, in keeping with myeloid changes being mainly associated with larger tumor burden. Not surprisingly, lymphoid:myeloid ratios decreased dramatically with tumor size/burden in all locations (Figure 1C).

Statistical comparisons of infiltrates in T1 and T2 tumor groups is complicated by their different sizes. Therefore, T1 tumors were grouped as being below and above the median size of 11.5 mm, which allowed large and small T1 tumors to be distinguished with 100% accuracy (95% cross-validated) by the combined differences in CD4/CD11b ratio, CD8/CD11b ratio, and PMN-MDSCs (*p* < 0.0001). Small T1 tumors could be compared with T2 tumors as they were the same size, and tests of equality of means showed only the CD4:CD11b ratio to be significantly different (*p* < 0.021). However, if the effect of this variable was statistically neutralized, the Treg content became significant (*p* < 0.004). These comparisons suggest that tumors self-regulate their immune content depending on their size, but the lymphoid compartment of T2 tumors was influenced by the presence of primary T1 tumors, especially with respect to Tregs. Clearly, tumor burden is a major influence on the intratumoral and systemic immune profiles in tumor-bearing animals, altering lymphoid and myeloid compartments, something that must be kept in mind when interpreting the effects of RT on immune parameters. Interestingly, tumor burden also correlated with the circulating neutrophil count and the circulating neutrophil:lymphocyte ratio, although no correlation was observed between tumor burden and the circulating lymphocyte count (Appendix A).

### 2.2. Radiation Plus Anti-PD-1 Monoclonal Antibody Can Drive Superior Systemic Tumor Control

Pilot experiments were conducted to develop the optimal hRT regimen for the treatment of established 6–8 mm diameter FSA1 T1 tumors in the absence of distant, microscopic disease. A single dose of 8 Gy caused tumor growth delay but no cure, 2 fractions of 8 Gy cured a few mice, while 8 Gy × 3 gave 100% cure (Figure 2A). We chose to use the 8 Gy × 3 regimen to the primary T1 tumors to determine if curative hRT could affect the growth of distant micrometastases, in the context of PD-1 checkpoint blockade (Figure 2B). T1 tumors were just palpable when T2 inocula were given and, in the face of the immunity present, only 35% of T2 (6 of 17) grew (Figure 2B left). hRT (8 Gy × 3) as a single modality, again, induced complete regression of T1 tumors, but any immunity that was generated was insufficient to cause abscopal regression of T2 tumors. Anti-PD-1 treatment alone, started on day 9, was ineffective both at controlling 6–8 mm primary tumors and at preventing the appearance of T2 tumors. However, addition of anti-PD-1 to focal hRT systemically controlled T2 as well as T1 tumors, curing 100% of mice (0 of 17 T2 grew, Figure 2B right, *p* = 0.018). Of note, radiation-induced lymphopenia was not altered by the addition of anti-PD1 (Appendix A).

### 2.3. hRT Alters the Immune Balance

The immune interplay between hRT-induced primary tumor regression and microscopic unirradiated deposits was examined, with and without anti-PD-1 treatment. Immune profiling was performed 7 days after hRT when irradiated primary tumors were regressing (Figure 3A). In the lymphoid compartment of irradiated T1 tumors, CD8+ T cells dramatically and significantly increased within the CD45+ population (*p* < 0.001), which was, to a lesser extent, also true in distant T2 tumors (*p* < 0.05) (Figure 3B). Although there was no increase in the overall proportion of CD8+ T cells that were positive for IFN-γ, the actual level of IFN-γ expression on a per cell basis was enhanced in T1 tumors and spleen, but less in T2 tumors. Tests of equality of group means showed that the increase in the number of CD8+ cells and IFN-γ expression levels in irradiated tumors were highly significant (*p* < 0.0001), and that either measure was 100% accurate in predicting treatment response. CD4+ cells increased only in T2 tumors after hRT, while Tregs increased in all sites, with statistical significance being reached in primary T1 tumors (*p* < 0.05). An increase myeloid cells was noticeable in the spleen, but the high level of M2 macrophages characteristic for primary tumors decreased dramatically after hRT (*p* < 0.01; Figure 3C). Lymphoid:myeloid ratios increased in T1 and T2 tumors, but for different reasons (i.e., CD8 vs. CD4, respectively) (Figure 3D). Tests of equality between means for T2 and irradiated T1 tumors showed that levels of IFN-γ expression could discriminate between groups with 92% accuracy. 

These profiles were consistent with radiation-induced regression of T1 tumors that was assisted by activated CD8+ cells, but this could not overcome systemic barriers to cause abscopal regression of microscopic lesions. This was confirmed by examining the PD-1/PD-L1 axis. Increases in CD8+ cells in the irradiated primary tumor were accompanied by increases in the number of CD8+ cells expressing PD1 and their levels of PD1 expression (Figure 3B). Similar increases in the number of PD-1+CD8+ T cells occurred in the spleen albeit without a change in expression levels. However, unirradiated T2 tumors if anything, surprisingly, had decreased PD-1+CD8+ numbers and expression levels. Similarly, the fraction of PD-L1+ myeloid cells and their expression levels were upregulated after hRT both in the irradiated primary tumor and in the spleen, but not in T2 tumors (Figure 3C). It should be noted that in this model hRT did not alter the, typically 60–80%, PD-L1 expression by tumor cells defined as CD45- cells in any of the tumor sites (Appendix A). 

Importantly, since the irradiated T1 tumors were regressing at the time of analysis, i.e., lesser tumor load in these animals, any changes observed in immune status could relate to (1) a direct radiation-immune effect, (2) an indirect effect through the reduction in tumor burden, or (3) both. However, the observation of the highly significant rise in the CD8+ infiltrate and their ability to produce IFN-γ in irradiated T1 tumors (Figure 3B) is independent of the reduction in tumor size (Figure 1A), as is the proportional rise in CD4 Tregs.

### 2.4. Tumor hRT Plus Systemic PD-1 Blockade Increases the Functionality of T Cells—Locally and Systemically

In contrast to hRT alone, combining systemic anti-PD1 with hRT triggered abscopal regression of microscopic disease (*p* = 0.018; Figure 2). Changes in immune cell composition in T1 tumors that were equal in size in these two groups and spleen were therefore examined. Anti-PD-1 alone had no effect on immune cell composition. Interestingly, compared to hRT alone, the combination of ICI and hRT further increased functionality of CD8+ T cells, measured by the intensity of the intracellular IFN-γ signal in both, the irradiated tumor (*p* = 0.003), and the spleen (*p* < 0.001; Figure 3B). T2 tumors obviously could not be evaluated as they did not grow.

## 3. Discussion

The immunogenic FSA1 murine fibrosarcoma used in this study has been the subject of many investigations by ourselves [7,14,15] and others [8,9,16]. In this study we show it to be PD-1 resistant, recapitulating the clinical response to anti-PD-1 described so far in STS patients. Two phase II studies of anti-PD-1 monotherapy in patients with advanced sarcoma have been published recently. One reported a lack of response to Nivolumab in patients with advanced uterine leiomyosarcoma [17], the other put the objective response rate in a STS cohort of various types receiving Pembrolizumab at 18% [18]. In our murine model, anti-PD-1 treatment was ineffective unless given with locally ablative hRT that, although it generated strong local CD8+ T cell responses, was itself remarkably inefficient in activating systemic immunity to cause abscopal regression even of microscopic disease. Importantly, this gain in anti-tumor efficacy with the hRT-ICI combo came at no apparent increased toxicity compared to either monotherapy.

In this study, T2 secondary inocula were given one week after the primary, which decreased the incidence of tumor take. Those that did grow presumably did so only because for them tolerance was induced. hRT (8 Gy × 3) monotherapy of T1 tumors dramatically enhanced the representation and function of intratumoral CD8+ T cells. This was also evident in the spleen, but T2 tumors seemed less affected. Indeed, in T2 tumors CD4+ cell changes were more marked than those for CD8+ cells. This study therefore suggests that hRT of primary tumors can activate immunity that is likely helpful in causing local regression, but that this is insufficient to break tolerance at other sites. This is not unlike what has been observed in the clinical setting. A comparison of paired of human sarcoma specimen before and after RT showed that radiation might promote an intratumoral immune effector signature and upregulate Major Histocompatibility Complex (MHC) class I expression [19]. Similarly, the expression of IFN-associated genes in post-SBRT tumor biopsy specimens correlated with an abscopal response in patients with advanced solid tumors treated with radiotherapy and pembrolizumab [20]. 

The general concept that hRT can interface with the immune system on multiple levels has been established. This includes tumor neoantigen release [21], tumor adjuvant release (DAMPS) [22] and enhanced type I IFN signaling [23] leading to activation of dendritic cells and the antigen processing machinery, activation and clonal expansion of effector T cells at the tumor site (TILs) and elsewhere [24,25], and tumor death receptor upregulation [26]. There is a growing body of evidence that ICIs can enhance many of these RT-induced effects, with preclinical and occasional clinical reports of abscopal responses [3,27]. The ideal radiation dose and fractionation regimen for stimulating a systemic anti-tumor immune response remains controversial. A dose per fraction of 8 Gy has been shown to best induce antitumor immunity and abscopal response in preclinical models [28,29,30] and hRT is being investigated with promising results. Timing is probably critical. In one study, combination immune-radiotherapy with concurrent but not sequential PD-1/PD-L1 signaling blockade improved treatment outcomes [31]. The Polish Sarcoma Study Group reported the results of 5 × 5 Gy followed by surgery three to seven days later. This strategy was found to be safe with an estimated 5-year local failure rate of 19% [32]. Furthermore, the preliminary results of an ongoing phase II trial (NCT02701153) on preoperative hRT (5 × 6 Gy) showed a 19% major wound complication rate and a mean pathologic necrosis score of 57%, which are similar to data from historical controls [33].

The considerable importance of tumor burden in determining the state of immunity has long been recognized, although it is only rarely taken into consideration, clinically or preclinically. Immunosuppression is integral to the proper function of both lymphoid and myeloid systems, with the former tending to show antigen specificity through Tregs, which often precedes a more general immune shutdown led by myeloid cells in patients with advanced disease [14]. The latter state may also allow metastases to develop. Experimental evidence over the past 50 years have documented many mechanisms of tumor escape [34], but the fact that both high and low sized inocula of FSA1, and other immunogenic tumors, can depress responses to subsequent challenges has been known for many years [7,35] and the importance of time between tumor challenges studied [36]. These experiments relate strongly to how tumor-induced changes in the immune system might promote the development of secondary tumors and present systemic barriers that have to be overcome before immune regression responses can be systemically generated. Not surprisingly, immunotherapy of FSA1 with C. parvum was only effective against moderate size inocula that could generate immunity [16]. Not only were small-size inocula not rejected but tumor take was actually enhanced by immunotherapy. These basic rules will apply to most immunogenic tumors and attempts at immunotherapy. 

## 4. Materials and Methods 

### 4.1. Mice

Animal experiments used immunocompetent 6- to 8-weeks-old C3Hf/Kam female mice bred and maintained in the defined-flora, AALAC-accredited animal facility of the Department of Radiation Oncology at the University of California, Los Angeles. All experimental protocols adhered to local and national animal care guidelines with IACUC approval. The protocol number for these experiments, it is: #1999-173-61, the approval date was 14 July 2017.

### 4.2. Cell Line and Reagents

FSA1 is a moderately immunogenic undifferentiated fibrosarcoma induced by single injection of methylcholanthrene into the flank of the same strain of C3H mice [37]. This tumor model has been used extensively, as quoted above. FSA1 cells were grown from frozen stock in vitro in Dulbecco Modified Eagle Medium (DMEM) (Corning Incorporated, Corning, NY, USA) supplemented with 10% heat-inactivated fetal bovine serum (FBS) (Sigma-Aldrich Corporation, St. Louis, MO, USA country) and 1% antibiotic-antimycotic (Corning Incorporated, Corning, NY, USA), at 37 °C in a humidified atmosphere containing 5% CO2 and harvested by trypsinization for injection. Anti-PD-1 mAb clone RMP1-14 were purchased from BioXCell^®^ (West Lebanon, NH, USA) and 300 µg per mouse injected intraperitoneally (i.p.) on days 9, 12, 15 and 18.

### 4.3. Tumor Model and RT

Tumor inocula of 5 × 10^5^ FSA1 cells in 100 µL of Dulbecco’s phosphate-buffered saline (DPBS; Corning Incorporated) were injected subcutaneously into the upper thigh region. The right thigh was used to generate primary tumors (T1) and seven days later, the same number of tumor cells was injected in the left side to mimic a metastatic site (T2). Mice were randomly assigned to one of four treatment groups with *n* = 17 mice per group: no treatment, anti-PD-1 monoclonal antibody (mAb) alone, RT alone, and the combination. RT was delivered to T1 tumors when they were 6–8 mm diameter, which was nine days after implantation i.e., two days after implantation of T2, when were not yet palpable, mimicking a micrometastatic site and based an average T1 tumor take rate of 100% (Figure 2). For hRT, mice were anesthetized via a single i.p. injection of ketamine 100 mg/kg and xylazine 6 mg/kg and 3 fractions of 8 Gy were delivered to T1 tumors using a Gulmay RS320 X-ray unit at 300 kV and 10 mA with 1.5 mm Cu and 3 mm Al beam filtration and a dose rate of 1.7 Gy/min (Gulmay Medical LtD., Camberley, Surrey, UK) with the rest of the body, including T2 tumors, shielded. Dosimetry was verified by TLDs and Gafchromic EBT3 film calibrated against NIST-traceable sources. Tumor diameters were measured in two perpendicular dimensions with Vernier calipers every third day for the duration of the experiment. Mice were euthanized if their body-conditioning score reached 2. Separate cohorts of *n* = 6 mice from each treatment group (plus *n* = 20 of untreated mice) were used for immune profiling to analyze treatment-related local and systemic immune changes. The data from the repeated experimental groups were consistent and therefore pooled for presentation. Mice were sacrificed on day 18 and tumor, spleen, and blood harvested for analysis.

### 4.4. Tissue Harvest and Immune Profiling

Complete differential blood counts were obtained from 100 µL whole blood taken via cardiac puncture of euthanized mice harvested into heparinized capillary tubes (Fisher Scientific, Pittsburg, PA, USA) using a Hemavet HV950 (Drew Scientific Inc., Miami Lakes, FL, USA). Tumors and spleens were harvested on ice. Tumors were cut into small pieces, enzymatically digested in 15 mL DPBS supplemented with Collagenase D 1 mg/mL (Sigma) and DNAse I 0.1 mg/mL (Sigma) for 30 min under slow rotation at 37 °C. Spleens were teased open and red blood cells lysed with ACK lysing buffer (Lonza) for 5 min. 1–2 × 10^6^ cells in DPBS were stained with fixable viability stain (BD Horizon FVS 510) prior to incubation with rat anti-mouse CD16/CD32 Fc blocking mAb (BD Pharmingen, San Jose, CA, USA; Clone 2.4G2) for 5 minutes at 4 °C followed by 30 mins at 4 °C with the following fluorochrome-conjugated antibodies in 2 separate panels: AF700 anti-mouse CD45.2 (BioLegend, San Diego, CA, USA; Clone 104), PerCP-Cy5.5 anti-mouse CD8a (BioLegend, Clone 53-6.7), APC/Cy7 anti-mouse CD4 (BioLegend, Clone GK1.5), PE anti-mouse CD25 PE (BioLegend, Clone PC61), APC anti-mouse FoxP3 (Invitrogen, Clone FJK-16s), BV785 anti-mouse PD-1 (BioLegend, Clone 29F.1A12), BV650 anti-mouse IFN-γ (BioLegend, Clone XMG1.2), FITC anti-mouse CD11b (BioLegend, Clone M1/70), APC/Cy7 anti-mouse Ly-6G (BioLegend, Clone 1A8), PE-CF594 anti-mouse Ly-6C (BD Horizon, Clone AL-21), APC anti-mouse F4/80 (BioLegend, Clone BM8), BV711 anti-mouse CD206 (BioLegend, Clone C068C2), PE anti-mouse PD-L1 (BioLegend, Clone 10F.9G2). For panels containing FoxP3, surface marker staining was performed first, prior to fixation and permeabilization using the BD Pharmingen™ Transcription-Factor Buffer Set. To determine the functionality of CD8+ T cells, single cell suspensions were stimulated ex vivo with PMA (Phorbol 12-Myristate 13-Acetate) and ionomycin (BD Pharmingen, Leukocyte Activation Cocktail with BD GolgiPlug™) for 6 h at 37 °C, prior to staining for surface markers and intracellular IFN-γ. Data were collected on a 14-color LSRFortessa™ (BD Biosciences, San Jose, CA, USA) and analyzed using FlowJo (FlowJo, LLC, Ashland, OR, USA). Gating strategies and subset definition for lymphoid and myeloid cell populations are shown in Appendix A and Appendix A. Local immune infiltrates referring to individual tumors are plotted against “tumor size” while systemic immune profiles (e.g., spleen) are shown versus collective “tumor burden” or “tumor load”, as the sum or T1 and T2.

### 4.5. Statistics

Data are presented as means +/− standard error of the mean (S.E.M.) using GraphPad Prism. Student’s *t*-tests were applied to compare differences between treatment groups, considered significant at the 5% probability level. Simple linear regression analysis was used to determine the correlation of each individual immune endpoint with tumor burden according to the Pearson correlation coefficient r, with multivariate regression analysis highlighting the strongest of these immune correlates. The nature of immune endpoints that were necessary and sufficient to distinguish between large and small tumors or between treatment groups (at equal tumor size) was established through multivariate linear discriminant analysis and accuracy measured with and without cross-validation. Differences in the proportion of T2 tumor control was assessed by the Fisher’s exact test. IBM SPSS Statistics for Windows, Version 24.0. (Armonk, NY, USA) was used to conduct the multiple regression and multivariate linear discriminant analysis. Heatmaps were constructed using R version 3.6.1.

## 5. Conclusions

In this study, anti-PD-1 ICI monotherapy was ineffective against palpable and microscopic tumors but could synergize if hRT was given to the primary supporting the notion that the PD-1/PD-L1 axis can be a key negative feedback mechanism. In other words, RT requires ICI to overcome the systemic barriers that are generated by primary tumors and that prevent the radiation-induced local immunity from spreading to generate effective abscopal regression of micrometastatic disease. This requirement is similar to the previous findings using anti-CTLA4 plus hRT [28]. This hypothesis is currently being tested in the SU2C-SARC032 randomized phase II trial (NCT03092323), comparing neoadjuvant RT with or without pembrolizumab followed by surgery in patients with clinically localized extremity STS at high risk for developing metastatic disease (tumor size > 5 cm, intermediate- to high-grade). It cautions that tumor burden and associated immune suppression are major factors in determining the outcome and immunologic correlative studies that examine functional endpoints will be critical for understanding of the such interactions [38].

## Figures and Tables

**Figure 1 cancers-12-00714-f001:**
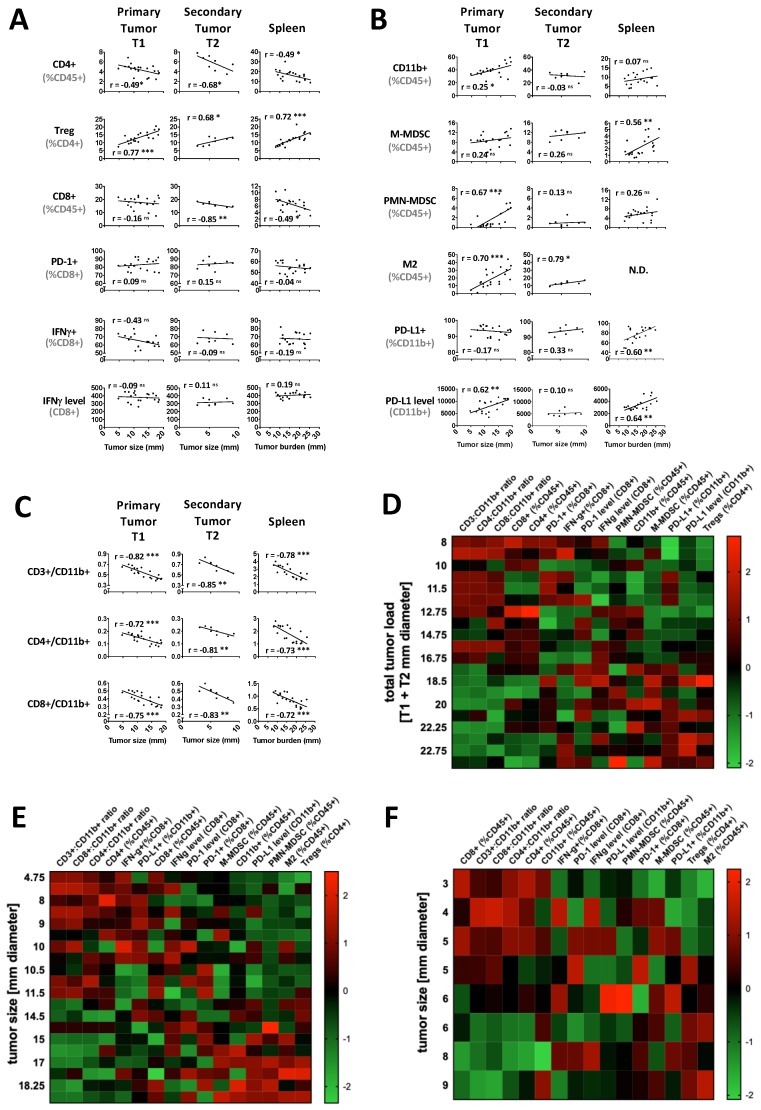
Immune suppressive mechanisms kick in locally and systemically as tumor load increases. The percentages of lymphoid (**A**) and myeloid immune cell subsets (**B**), and the ratios thereof (**C**) in the primary tumor T1, secondary tumor T2 and spleen of untreated mice depend on the tumor size and total tumor burden (*n* = 20). Heatmap showing a rise in myeloid and lymphoid suppressor cells dominance with increase in tumor burden in spleen (**D**), in T1 (**E**), and in T2 (**F**). Heatmap was constructed by normalizing each immune cell subset across all mice. Values are *z*-scores for each immune subset in each mouse. r represents the Pearson correlation coefficient. * *p* < 0.05; ** *p* < 0.01; *** *p* < 0.001; ns not significant.

**Figure 2 cancers-12-00714-f002:**
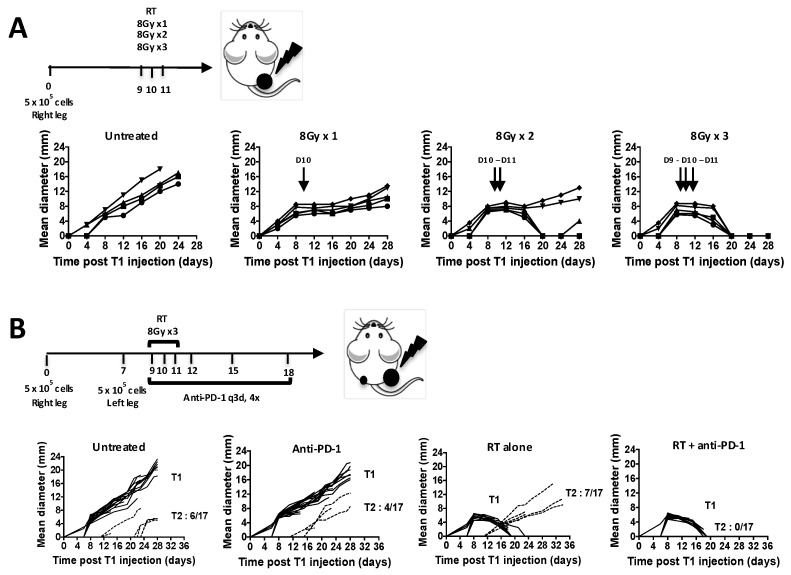
Micrometastatic fibrosarcoma is effectively controlled with ablative irradiation combined with anti-PD-1. (**A**) In a preliminary experiment, unilateral tumors were treated with irradiation (RT) alone to analyze the radiation dose response (*n* = 5 per radiation dose). (**B**) Primary (T1) and secondary (T2) tumors diameters were monitored in the no treatment, anti-PD-1, irradiation and combination groups (*n* = 17 per treatment group).

**Figure 3 cancers-12-00714-f003:**
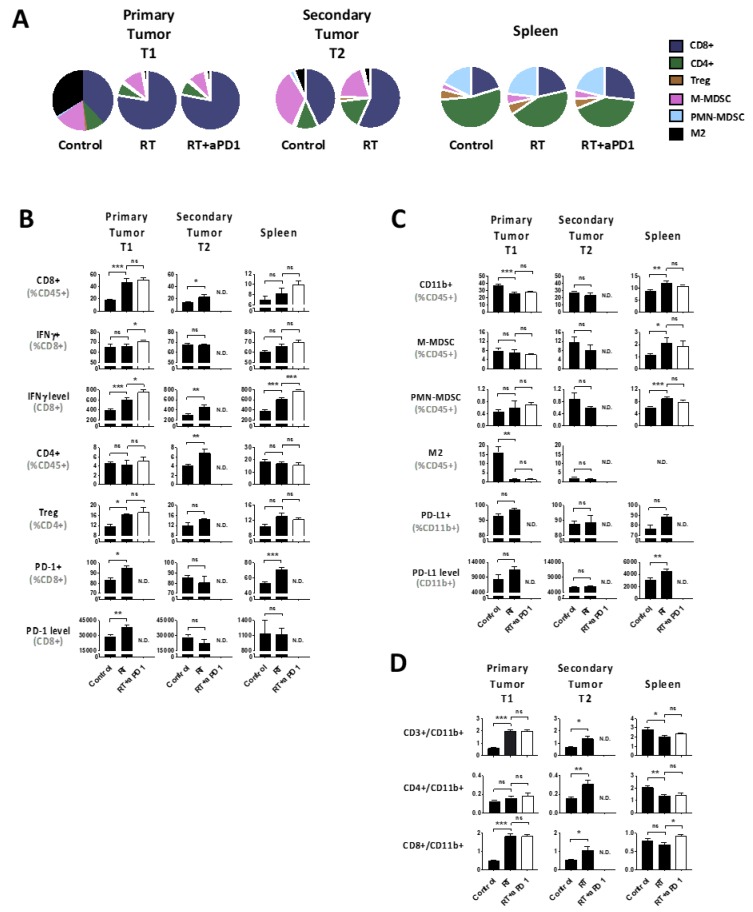
Immune cell profiles and PD-1/PD-L1 expression levels in the primary tumor T1, the secondary tumor T2 and the spleen in response to radiation therapy (RT) with or without anti-PD-1 (RT+aPD1) (*n* = 6 per group). (**A**) Overall mean relative changes in immune cells subsets; (**B**) Lymphoid subsets; (**C**) Myeloid subsets; (**D**) Ratio of lymphoid-to-myeloid subsets. In every tissue sample, there was a proportion of undetermined cells among the CD45+ cells, which did not change between the control and the irradiated groups (not shown). Data are mean +/− S.E.M. with * *p* < 0.05; ** *p* < 0.01; *** *p* < 0.001; ns not significant. ND: No Data.

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
