# Peer review of "Tumor Size Matters—Understanding Concomitant Tumor Immunity in the Context of Hypofractionated Radiotherapy with Immunotherapy"

_cancers, 2020, doi:10.3390/cancers12030714_

Round 1
Reviewer 1 Report
The Authors study immune-profiles in spleen and macroscopic tumors harvested from bilateral syngeneic FSA1 fibrosarcoma murine model before and after radio- and anti-PD-1 therapies, alone or in combination. The aim of the study is to decipher what is the most important axis for synergy between those two therapies. This study is important, because there is an increasing recognition (but still limited understanding) that RT can synergize with immunotherapies, significantly improving therapeutic outcomes (what is tested in many currently ongoing clinical trials). Whereas, I find the results presented in the study interesting, well presented and worth publishing, I’m not certain about putting so much attention to the tumor size in the title of the article and conclusions (correlation is not causation). Especially to the statements that tumor burden is a major confounding factor in immune analysis, as this hasn’t been shown. There are various effects at play, but it seems more likely that it is the immune profile inside the tumor driving its growth and only to limited extent it is happening the other way around (through concomitant immunity). In other words, when the tumor immune composition is known (e.g. from biopsies/tissue samples), then the burden might not be a confounding factor essential for the outcome analysis.
Other comments:
- I understand that in the second column of panels A-C in Fig 1. we see the scatter plots only for those T2 tumors that grew successfully. If that is the case shouldn’t we use only those tumors in combination with T1 size for column 3, i.e. for tumor burden? Otherwise we add 0 as T2 to the tumor burden, which is not completely accurate.
- Same as above to panels D-E.
- I think it might be interesting to see the comparison of mean/medians (boxplots) comparing various variables for T1 tumors for which T2 tumor grew successfully with those for which T2 was not palpable.
- Line 97: less significant -> not significant in the spleen
- Please use different color pallets for Fig. 1D-F and Fig. 3 A, i.e. use pallets that are optimized for black and white printouts. Now greyscale printout is unreadable.
- It seems that Fig. 3 panel A is not referenced in the text. Some statistics for this panel showing if distribution changed before vs after would be nice.
Reviewer 2 Report
Re: review
“Tumor Size Matters - Understanding Concomitant Tumor Immunity in the Context of Hypofractionated Radiotherapy with Immunotherapy” – Cancers
The purpose of this study was to evaluate the kinetics of different immune cell populations after hypofractionated radiotherapy (hRT) and PD-1 targeted therapy in a mouse syngeneic fibrosarcoma model and their impact on mediating abscopal effect. While this approach may be reasonable and potentially beneficial for cancer research, this report suffers from a number of weaknesses, as described below.
The immune cells investigation to determine the impact of treatment in mediating “abscopal” regression was done only one time using 6 mice per group. To validate their conclusion at least one more experiment in the same setting must be done. They used through all paper only one method, flow cytometry. It is unusual to not validate the conclusion using a second method of validation.
Figure 2B raises some interesting questions. Despite having no effect on primary tumor (T1), anti-PD-1 appears to have a strong effect on secondary tumor (T2) development. 4 out of 17 mice developed T2 as compared to hRT group alone where 7 out of mice developed T2. If we follow the rational of using hRT for inducing asbscopal effect, than RT group should had much less T2 development than untreated control and anti-PD-1 groups alone. Moreover, the untreated group showed T2 development in only 6 out of 17 mice. Taken together, my question is T2 development is just a random effect due to troubleshooting with tumor inoculation? I do realize that the “clinical” observation experiment with 17 mice per group where only growth curves are investigated may be enough to justify doing one experiment. However, having this trend with less T2 development in untreated and anti-PD-1 groups, the experiment should be repeated. Interestingly, they have the parallel experiment where they do the immune cells investigation, but growth curves are not shown. Why? Also, why the mice from the big experiment were not investigated for immune cell populations.
Data for anti-PD-1 group are not shown in figures 3 and 4. Unusual for a comparison between treatments groups (especially knowing that anti-PD-1 showed the best anti-T2 effect as compared to RT alone and untreated groups). Also I do not understand why figure 3 shows comparison between untreated control and RT and figure 4 shows the RT data vs. RT+anti+PD-1. These should be one figure, not 2 (again unusual way to presents scientific data by a clear comparison between the immune cell populations from all 4 groups). Moreover, figure 4 does not show the comparison between RT and RT+anti-PD-1 for PD1 (%CD8+), PD-1 level (CD11b+) in lymphoid subsets and PD-L1 (%CD11b+) and PD-1 level (CD11b+) in myeloid subsets. Data for these information are shown in figure 3 whet RT was compared to untreated. Any reason?
PD-L1 expression on fibrosarcoma cell line is missing. They used an anti-PD-1 treatment. This mainly works by blocking PD-1/PD-L1 axis. PD-L1 data in tumor must be shown. There are very few studies in humans which indeed show some benefits of anti-PD-1 therapy in patients with negative PD-L1. These were mainly in the context of NK cells population. Speaking about NK, any particular reason for not showing NK and M1 cells data?
The design of the ex vivo with PMA and ionomycin is not appropriate for the purpose of the study. Pure CD8+ T cells enriched from spleen should have been used, not the whole population (not clear the source, blood, spleen?). What does “functionality of CD8+ T cells” mean in the context of the model? If they used the whole immune cells population, PMA and ionomcyin will stimulate many other cells type beside CD8+ T cells. Considering that there are big differences in the percentages of subets between groups, this non-specific stimulation in their experimental setting cannot prove the functionality of T cells. Moreover, irradiated fibrosarcoma cells or lyaste and not PMA/ionomycin should have been used for specific stimulation.
Authors discuss that combination of hRT and anti-PD-1 induced complete abscopal regression. By definition, abscopal effect is mediated only by a local treatment which generates a systemic anti-tumor effect. Anti PD-1 is a systemic treatment which “manipulates” immune system. Also, they use a very early tumor model for secondary T2 tumor. It looks like their combination does not induce an “abscopal” regression; it appears that the treatment prevent tumor development. Discussion must be written accordingly to not bias the readers.
What clone of anti-PD1 was used? It is mentioned only the company. So far I know, BioXCell produces anti-mouse PD-1 MAbs in rats and armenian hamsters. If the antibody used originated from one of these species what was the rational for the last dose. Nine days (time between first and fourth anti-PD-1 doses) is enough time to mount an antibody response by host (mouse) against the Fv part of the foreign antibody (rat or hamster), practically neutralizing the circulation MAbs and the fourth dose.
The authors use only one tumor model of fibrosarcoma. So far I know “Cancers” journal requires mandatory for pre-clinical manuscripts the use of at least two cell lines. Thus, the authors should present data using at least two tumor models. Using only one cell line raises the question (always in literature): Are their results determined by the effect of treatment against fibrosarcoma or against that particular cell line used?
This manuscript is supposed to address the topic "Animal Models for Radiotherapy Research". I am not sure what does the editor mean but the model used is commonly used in Radiation Oncology research. Which is their novelty with regard to animal model?
Each figure should have a clear presentation of the experiment with number of mice and therapy used. For example, it is not clear how many mice were used for the first experiment (Figure 1). Growth curves should be also shown.
Taken together, all the above potential biases not repeating the flow cytometry investigation for all 4 groups comparison significantly limit the interpretation of the findings and challenge the validity of the conclusions made. Thus, a resubmitted manuscript is needed. This manuscript cannot be accepted without major revisions and thorough new experiments.
Thank you.
Round 2
Reviewer 2 Report
I would like to thank the authors for their remarkable work to answer to reviewers’ questions in such a short time. The revised manuscript is responsive to previous review comments including adding new figures, discussion, and rephrasing some sentences. I sincerely believe that the updated manuscript is suitable for publication now.
Thank you.